# Simultaneous Determination of 21 Sulfonamides in Poultry Eggs Using Ionic Liquid-Modified Molecularly Imprinted Polymer SPE and UPLC–MS/MS

**DOI:** 10.3390/molecules27154953

**Published:** 2022-08-04

**Authors:** Decheng Suo, Su Zhang, Zhanteng Song, Shi Wang, Yang Li, Xia Fan

**Affiliations:** 1Institute of Quality Standards and Testing Technology for Agricultural Products, Chinese Academy of Agricultural Science, Beijing 100081, China; 2Institute of Quality Standards and Testing Technology for Agricultural Products, Xinjiang Academy of Agricultural Science, Urumqi 830091, China

**Keywords:** sulfonamides, ionic liquids, molecularly imprinted, poultry eggs, UPLC–MS/MS

## Abstract

An ionic liquid-modified molecularly imprinted polymer (IL-MIP) composite with sulfamethazine as a template molecule and methyl acrylic acid and 1-aminopropyl-3-methylimidazolium bromide as functional monomers was successfully synthesized. The achieved IL-MIP was characterized and evaluated in detail and utilized in the extraction and cleanup of sulfonamides (SAs) in poultry egg samples. The results demonstrated that the IL-MIP possessed a broad reorganization toward SAs and could selectively adsorb 21 kinds of SA compounds. Furthermore, the solid-phase extraction column based on the IL-MIP was used in the extraction and cleanup of 21 SAs in eggs, and the confirmatory detection of SAs was performed using ultraperformance liquid chromatography–tandem mass spectrometry. Under optimum conditions, the limits of detection (LODs) for all SAs ranged from 0.1 ng·g^−1^ to 1.5 ng·g^−1^, and the LOD of this method was better than those of the existing methods. The recoveries of SA compounds spiked in egg samples ranged from 84.3% to 105.8%, with low relative standard deviations (<15%). The developed method based on the IL-MIP extraction and cleanup was successfully used in the detection of 21 SAs in more than 100 real poultry egg samples. The results indicated that the proposed method was suitable for detecting 21 SAs in poultry eggs.

## 1. Introduction

Sulfonamides (SAs) are a group of compounds with a *p*-aminobenzenesulfonamide structure and are used to prevent and treat bacterial infectious diseases [1]. These drugs are commonly utilized as veterinary antibiotics and have a broad spectrum of antimicrobial activity. SAs, which are administered orally, are absorbed rapidly, highly stable, and do not deteriorate easily [2,3]. However, the application of SA compounds in a farm livestock can result in residues in products originating from animals. These residues, when present in animal food, considerably harm human health [4,5,6]. Various SA-resistant strains have been identified in animals, and these strains can be transferred from animals to humans through food and environment by animal waste [7,8,9]. Considering the risk of toxicity and antibiotic resistance, China and the European Union have limited the maximum residue limits of SAs in animal-derived food to 100 μg·kg^−1^ [10]. Moreover, the use of SAs as feed additives or veterinary drugs are prohibited during the laying period of poultry. Moreover, the principle of “zero tolerance” is applied to eggs [11]. Therefore, simple and reliable analytical methods are required to meet the requirements and ensure the safety of consumer eggs.

To date, many analytical methods have been developed to detect SA residues in eggs [12,13,14,15,16,17,18,19,20,21,22,23,24]. However, these methods have no specificity for SAs, and usually extract some impurities, such as cholesterol and phospholipids, which will affect the accuracy of SA determination. Additional pretreatment steps are required to remove interference. Therefore, an effective and simply sample preparation technology should be developed to eliminate impurities and ensure the accuracy of the analytical methods.

The molecular imprinting technology involves the preparation of polymers by preassembly and crosslinking. Target molecules and functional monomers are copolymerized through covalent or noncovalent interactions [25]. This method can recognize targets in complex samples with high selectivity. Currently, molecular imprinted polymers (MIPs) have been applied successfully to analyze SAs [26,27,28]. MIPs with sulfamethazine (SMZ) as the template have been synthesized on the surface of silica gel. The MIPs have been applied to determine sulfathiazole, SMZ, and sulfamerazine in pork and chicken muscles [27]. The water-compatible sulfadiazine@MIP with surface-grafted poly(2-hydroxyethyl methacrylate) has been synthesized through one-pot reactions and applied in solid-phase extraction (SPE) to detect six SAs in muscle and water [26]. A magnetic MIP with ZnO@CF was developed for the extraction and enrichment of SMZ, sulfadiazine, and sulfadimethoxine (SDM) in milk and water samples [28]. These SAs are selectively isolated, and matrix interferences are eliminated using these polymers. However, these MIPs only selectively purify 3–7 SAs. The adsorption capacity (Q) of these polymers cannot meet the requirements of high throughput and low cost for the detection of multiple SAs.

Ionic liquids (ILs) belong to a group of salts consisting of a large organic cation and a small organic or inorganic anion. ILs are green solvents characterized by relatively low melting points because of the inefficient packing of bulky cations and small anions. These salts, as monomers and additives, have been used to prepare MIPs [29,30,31,32,33]. MIPs based on ILs have been applied to detect 6-benzylaminopurine in food and water [30], dicofol in celery [31], and *p*-nitroaniline in wastewater [32]. Notably, the IL 1-allyl-3-vinylimidazolium has been used as a functional monomer to synthesize MIP on the surface of silica carriers. This MIP has excellent selectivity toward three SA antibiotics (i.e., sulfamonomethoxine, sulfamethoxazole, and sulfadiazine) in methanol (MeOH) solution [33].

In this study, several IL-MIPs have been synthesized through precipitation polymerization by using various ILs as functional monomers. The adsorption properties of these IL-MIPs are studied; the most effective IL-MIP with 1-aminopropyl-3-methylimidazole bromide (AMB) as monomer (MMIP) is selected as the SPE material to extract and clean 21 SAs in egg samples. A novel analytical method combined with ultraperformance LC–MS/MS (UPLC–MS/MS) has been developed to detect multiple residues of SAs in poultry eggs.

## 2. Materials and Methods

### 2.1. Materials and Reagents

Formic acid (FA) for MS and acetonitrile (ACN) and MeOH for LC were purchased from Fisher Technology, Inc. (Muskegon, MI, USA). SMZ was obtained from Sigma-Aldrich (St. Louis, MO, USA). Water was purified using the Milli-Q water purification system from Millipore (Bedford, MA, USA). AMB was ordered from Innochem Technology Co., Ltd. (Beijing, China). Butyl-3-methylimidazole bromide (BMB) was purchased from Yuanye Biotechnology Co., Ltd. (Shanghai, China). 1-Allyl-3-vinylimidazolium bromide (AVB) and 2,1-ethyl-3-methylimidazole bromide were obtained from Chengjie Chem. Co. Ltd. (Shanghai, China). Methacryclic acid (MAA), 2,2-azodiisobutyronitrile (AIBN), and ethylene glycol dimethacrylate (EGMA) were acquired from Tokyo Chemical Industry (Tokyo, Japan). Sulfadiazine, sulfamethoxazole, sulfathiazole, sulfamerazine, sulfafurazole, sulfadimethoxine, sulfadimoxine, sulfamethizole, sulfabenzamide, sulfisomidine, SMZ, sulfametoxydiazine, sulfamethoxypyridazine, sulfamonomethoxine, sulfachloropyridazine, sulfachloropyrazine, sulfaquinoxaline, sulfanitran, sulfaphenazole, sulfapyrazole, and sulfisoxazole were purchased from Sigma (St. Louis, MO, USA), Dr. Ehrenstorfer (Augsburg, Germany), and WITEGA Laboratories (Berlin, Germany). Twenty SA internal standard (IS) solutions (BePure^®^) in MeOH were purchased from Manhage Bio-technology company (Beijing, China).

About 5 mg SA was weighed accurately and used to prepare a 100 μg·mL^−1^ standard solution with MeOH. Standard working solutions with standard working series concentrations were diluted with 0.1% FA in water/MeOH (90:10, *v/v*). A total of 100 egg samples were acquired from different markets and poultry farms. Positive samples for the control group were obtained from animal experiments [34].

### 2.2. Method

#### 2.2.1. Preparation of IL-MIP

In brief, 1 mmol SMZ, 4 mmol MAA, and 4 mmol IL (AMB) were added to 20 mL ACN + water (1 + 1) in a 40 mL sealed glass tube and dissolved by sonication for 20 min and mixing for 2 h after nitrogen filling. Then, 20 mmol EGMA and 50 mg AIBN were added for the polymerization reaction. The mixture was maintained for 5 min by a nitrogen filling. The mixture was incubated in a shaking water bath at 60 °C for 12 h, and the products of solids were recovered by filtration. The products were washed using 50 mL MeOH + acetic acid (9:1, *v/**v*) solution for 24 h by Soxhlet extraction, and dried in an oven at 60 °C for 12 h. The newly synthesized IL-MIPs (MMIP) were subsequently placed in a glass bottle at room temperature for further use. Other MIPs were prepared similarly but without the template molecule (MNIP); without IL (NMIP); and AVB (VMIP), BMB (BMIP), and 1-ethyl-3-methylimidazole bromide (EMIP) as ILs.

The MMIP SPE column was prepared as follows: a piece of polyethylene sieve plate was inserted at the bottom of the column tube of a blank SPE column. Then, 100 mg MMIP was placed into the column and gently tapped to make the content compact. The polymer was plugged with a sieve plate. MeOH and water were made to pass through the column, and possible impurities were removed and solvated.

#### 2.2.2. Sample Preparation

The extraction solution was derived in accordance with a previously published method [16,35,36,37]. Exactly 5 g of the sample was weighed in a 50 mL stoppered plastic tube, added successively with 10 µL IS solution and 10 mL of 20 mM phosphate buffer saline (PBS) solution (pH 3.0), and subjected to vortex mixing for 30 s. The mixture was oscillated for 10 min in an oscillator and filtered by a quantitative filter paper. Then, 5 mL of the supernatant was prepared for MIP purification [38].

The solution was loaded into the MIP column at a low flow rate. Afterwards, the column was washed with 3.0 mL water and vacuum dried for 2 min. The column was eluted with 3.0 mL MeOH containing 5% acetic acid. The eluate was evaporated to dryness under a stream of nitrogen at 40 °C and reconstituted in 0.5 mL of 0.1% FA in water/ME (90:10, *v/**v*). Then, the mixture was filtered using a 0.22 µm filter membrane and transferred into vials prior to the LC–MS/MS.

#### 2.2.3. Instrumentation

SAs were detected in accordance with a previously described method (Suo et al., 2019) [34]. The analytes were separated using the Waters Acquity UPLC coupled with the CSH C18 column (150 mm length × 3.0 mm internal diameter; 1.7 µm particle size) and detected using the Xevo TQS MS/MS (Milford, MA, USA). The mobile phase flow rate was 0.3 mL·min^−1^, and the following linear gradient was used. The conditions of the mobile phase and the mass spectrum are shown in Appendix A. The Masslynx V4.1 software was used for instrument controlling and data processing (Waters, MA, USA). The extracted ion chromatograms of 21 SAs are shown in Figure 1.

## 3. Results and Discussion

### 3.1. Selection of IL

Many ILs have been applied as a monomer or additive to synthesize IL-MIPs [29,30,31,32,33]. In these cases, MAA is an acidic monomer and can form strong interactions with ILs and target molecules. The best adsorption has been obtained at a molar ratio of 1:1 [31]. Mixtures of four specific ILs and MAA were used as functional monomers in the current work to prepare MIPs (molar ratio of 1:1) and study the functional monomer. Polymer materials were successfully synthesized. The values of sorbents (Q value) for SMZ were evaluated using the equation reported in previous studies [30,39,40,41]. The results are shown in Appendix A. Compared with other ILs, AMB was the most efficient functional monomer for SMZ because of the unique function of its amino propyl group, which improved interactions between the functional monomer and SMZ.

### 3.2. Evaluation of IL-MIPs

#### 3.2.1. Characteristics of MIPs

Fourier transform infrared (FT-IR) spectrometry (VERTEX70) was used to obtain the FT-IR spectra of MMIP, NMIP, and MNIP (Appendix A). These materials had similar spectra, indicating that the template molecules were completely washed away. The spectra of MMIP and MNIP showed imidazole ring vibrations at approximately 956, 879, and 755 cm^−1^. Stretching vibrations introduced by IL also appeared in the spectra of MMIP and MNIP. These additional peaks indicated that the IL was covalently bonded to MIPs.

The morphologies of MMIP, NMIP, and MNIP were analyzed through scanning electron microscopy (Figure 2). MMIP had higher aggregation and adhesion than NMIP because of the high viscosity of IL. MMIP and MNIP were spherical and had diameters of 1 µm, which was higher than that of NMIP because of IL. MMIP showed an irregular distribution compared with MNIP and NMIP because of the addition and washing out of the imprinted template of MMIP.

#### 3.2.2. Selectivity and Specificity of MMIP and Other MIPs

MMIP has been proven to be the most effective material for SMZ (Appendix A), but the adsorption effect of other SAs of MMIP has not been evaluated. In addition, no significant difference was found in the adsorption effect of several SAs, such as sulfaquinoxaline, with or without a template in IL-MIPs. Therefore, the adsorption properties of these MIPs for 21 SAs were further studied to evaluate selectivity and specificity. Six types of MIP (50 mg) were individually added to each plastic pipe with 2.0 mL PBS solution (pH 3.4) containing 200 ng·mL^−1^ of 21 SAs. After shaking the mixture for 1 h, the suspension was filtered using a 0.22 μm membrane and analyzed via LC–MS/MS. No MIP was added as a control sample. Table 1 shows the adsorption percentages of 21 SAs. The results showed that 21 SAs had good retention with MMIP (above 60%). The adsorption percentages of several SAs containing pyrazole groups, such as sulfisoxazole and sulfaphenazole, were 62.7–75.4%. This result could be attributed to structural differences and SMZ.

Evident differences were observed between the Q values of different types of MIPs and each SA. Several SAs, such as SMZ, exhibited remarkable differences in Q values between MMIP and MNIP. Few gaps were found between MMIP and NMIP or different IL-MIPs. The adsorption factors of these drugs were possibly based on the mechanism of the molecular imprinting adsorption. However, sulfachloropyrazine, sulfaquinoxaline, sulfamonomethoxine, sulfametoxydiazine, and sulfamerazine showed low deviation between MMIP and MNIP and high deviation between MMIP and NMIP. The force related to molecular imprinting might not be the main factor for adsorption. The ionic bond between the IL and sulfa group might play a major role in the adsorption of these drugs. Several SAs might have distinct Q values in different IL-MIPs because of the diverse substituent groups of ILs. IL-MIP had higher Q than MNIP. Evident differences were found among MMIP, NMIP, and IL–MIP. Hence, SA adsorption by MMIP might involve a complex mechanism probably because of the interactions between the analyte and ILs or the anion exchange polymer-confined ILs. All MIPs showed good adsorption for sulfanitran. Hence, other adsorption mechanisms might be involved with sulfanitran. Four types of IL-MIPs (MMIP [21], VMIP [14], BMIP [11], and EMIP [13]) had different numbers of SAs with recovery rates higher than 50%. Although other IL-MIPs had good adsorption for several SAs, MMIP was the best choice for 21 SAs in eggs. Compared with other ILs, ABM exhibited better performance, which might be due to the π–π and ion interactions between the analyte group and imidazole of specific ILs. Another factor was the hydrogen and amino bond interaction between the amino propyl group of specific ILs and group of SAs.

### 3.3. Optimization of SPE Purification Procedures

Several parameters of MIP SPE were studied prior to application by using a control solution to evaluate the cleanup and performance of MMIP. These parameters included the pH of PBS, the amount of MMIP, and the type and volume of the elution solvent.

Batch adsorption studies were conducted at a pH range of 3.0–10.0 and 25 °C to show the pH effect on the adsorption of SAs onto MMIP. The results showed that PBS with different pH values had no evident difference on the Q of MMIP (results not shown). Therefore, the extraction solvent did not need additional pH adjustment.

The effects of different amounts of MMIP (i.e., 20, 50, 100, 200, and 500 mg) applied to the SPE column were analyzed. The extraction solvent (5 mL) from a 5.0 g egg sample with 100 ng·mL^−1^ SAs was loaded to the SPE column. The results indicated that 100 mg polymers were adequate, and satisfactory recoveries were obtained. Further increments in the amount of MMIP did not significantly improve the recoveries of SAs. Therefore, the dosage of MMIP material was set to 100 mg.

Different proportions of acidified MeOH and ACN were tested as elution solvents to optimize the elution conditions and obtain the highest recoveries of SAs, MeOH, and ACN (Appendix A). Poor recoveries for several SAs (below 50%) were obtained using MeOH and ACN. Satisfactory recovery was obtained using a mixture of MeOH/acetic acid (95:5, *v/v*) as the elution solvent. Moreover, the volume of the elution solvent (i.e., 1–5 mL) was studied, and 3 mL of 5% acetic acid in MeOH was selected as the eluting solvent to elute the target analytes fully.

### 3.4. Validation of the Proposed Method

Method specificity was evaluated by analyzing blank egg samples from different sources. No interfering peak was observed at the time of analyses.

The detection limit (LOD) was determined in accordance with the signal-to-noise ratio (S/N) of the MRM chromatographic peak of 10 blank samples, which was higher than 3. LOQ, which was higher than 10, was determined in accordance with S/N. The results are shown in Table 2. The mixed standard solutions of SAs and 18 IS were added to prepare a series of solutions with concentrations of 1, 2, 5, 10, 20, 50, 100, and 200 µg·L^−1^ SAs and 10 µg·L^−1^ IS. Calibration curves were calculated using the ratio of each SA (the specific concentration range is shown in Table 2) and 10 ng·mL^−1^ of the corresponding IS. The linear regression equation and correlation coefficient (Table 2) revealed that the 21 target compounds had good linear relationships and that the correlation coefficient was not less than 0.990.

Experimental accuracy was determined by spiking blank samples with 21 SAs at three concentration levels (i.e., 2, 10, and 100 ng·g^−1^). Six replicates of each were analyzed continuously for three days. The results are summarized in Table 3. The average recovery ranged from 84.3% to 105.8% and the relative standard deviation (RSD) was <15%.

### 3.5. Evaluation of the SPE Reusability

Reusability is the main advantage of MIP. To evaluate the reusability of IL-MIP SPE, we assessed the adsorption–regeneration cycles of the SPE column on the egg extraction solution. These SPE columns, which were used on the PBS and egg extraction solution, were collected. The SPE columns were followed by 5 mL MeOH + acetic acid (9:1, *v/v*) washed at room temperature to neutralize the polymers. The retreated SPE columns were used to detect SMZ. To evaluate the reusability of the imprinted polymers, we performed adsorption–regeneration cycles up to 10 times, and the relative adsorption of SMZ was adopted as the evaluation parameter (Appendix A). The relative adsorption of the egg extraction solution was below 90% after five times of recycling and below 80% after seven times of recycling. This result was achieved possibly because the ester and unknown substance in the egg extraction solution affected the performance of SPE columns after repeated use. Despite these considerations, the SPE columns were reused five times for sample analysis.

### 3.6. Comparison with a Reported Method

Several studies determined SAs in eggs. Compared with other analytical methods shown in Table 4, the proposed method provided a simpler and faster means of extracting SA, lower LOD, higher recovery, and better precision. In addition, the proposed technique required no additional organic solvent consumption.

### 3.7. Application and Confirmation

The developed method was used to determine SAs in 100 egg and 3 positive egg samples from an animal study for further validation [34]. Positive results were analyzed in accordance with the standard method of China and this method (SN/T 4057-2014) [15]. Positive results (Figure 3) indicated that SAs were detected in 4 out of 100 samples. Three types of SAs were positively determined in egg samples, and the measured concentrations were 50 and 340 µg/kg for SMZ and 123 and 530 µg/kg for sulfaquinoxaline and sulfadiazine, respectively. Good contrasting results (RSD < 15%) were obtained using the developed method. This finding agreed with the confirmatory determination. A small amount of SMZ, which was not found with SN/T 4057-2014, was detected using the proposed method in eggs collected at 14 days after drug withdrawal (1.8 ± 0.3 µg/kg).

## 4. Conclusions

MMIP was synthesized by AMB and MMA as mixed functional monomers for the selective separation of 21 SAs in egg extracts. MMIP was prepared via precipitation polymerization by using SMZ as a template and EGMA as a crosslinker. A method for the simultaneous determination of 21 SAs in poultry eggs with IL-MIP was also developed. The proposed method provided a practical, simple, fast, and environmentally friendly approach of analyzing 21 SAs in poultry eggs.

## Figures and Tables

**Figure 1 molecules-27-04953-f001:**
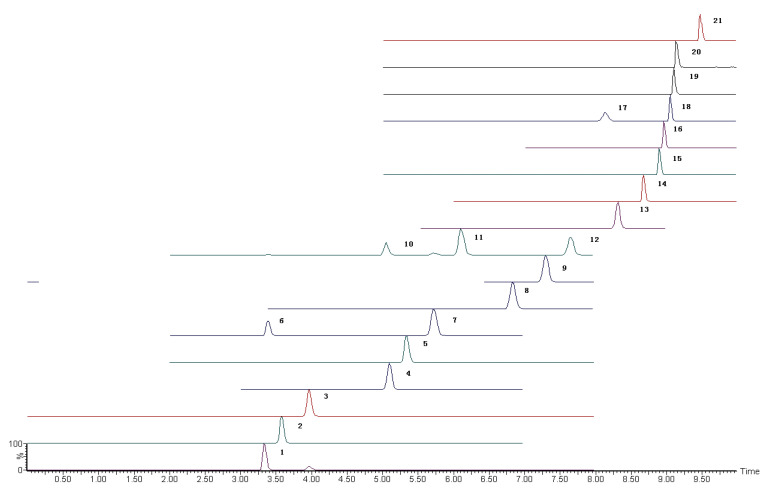
The extracted ion chromatograms of 21 SAs. 1. Sulfisomidine; 2. sulfametoxydiazine; 3. sulfadiazine; 4. sulfathiazole; 5. sulfachloropyridazine; 6. sulfamerazine; 7. sulfafurazole; 8. sulfamethizole; 9. sulfamethoxazole; 10. sulfamethazine; 11. sulfamethoxypyridazine; 12. sulfamonomethoxine; 13. sulfachloropyrazine; 14. sulfadoxine; 15. sulfisoxazole; 16. sulfabenzamidec; 17. sulfaphenazole; 18. sulfadimethoxine; 19. sulfapyrazole; 20. sulfaquinoxaline; 21. sulfanitran.

**Figure 2 molecules-27-04953-f002:**
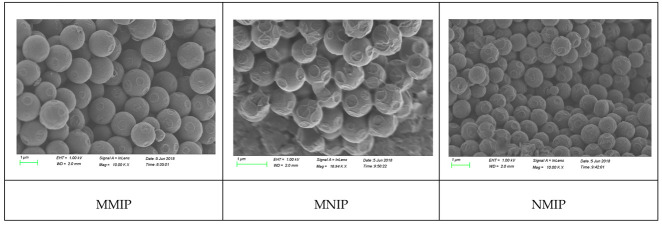
The SEM spectrum of MIP.

**Figure 3 molecules-27-04953-f003:**
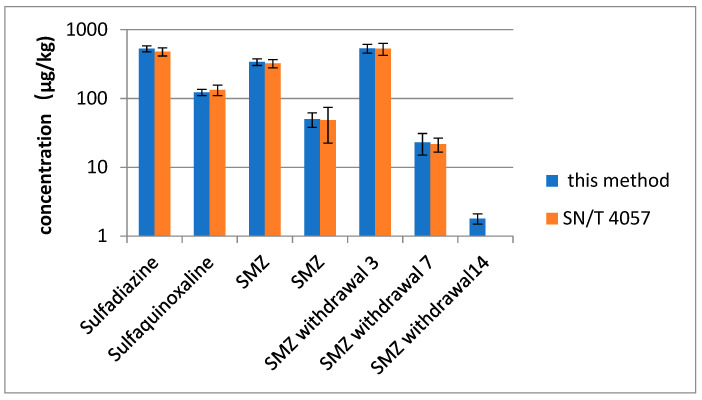
The result of SAs in egg samples with this method and SN/4057. SMZ withdrawal 3, 7, 14 is egg samples from 3, 7, 14 days after drug withdrawal.

**Table 1 molecules-27-04953-t001:** Adsorption rates of 6 kinds of MIP.

Drug	Adsorption Rate (%)					
	MMIP	MNIP	NMIP	VMIP	BMIP	EMIP
Sulfadiazine	80 ± 0.9	21 ± 71.5	74.4 ± 2.8	83.6 ± 2.3	72.5 ± 2.8	74.1 ± 16.2
Sulfamethoxazole	60.7 ± 10.5	41.2 ± 8.3	26.3 ± 11.1	58.4 ± 9.6	42.3 ± 8.7	63.5 ± 11.6
Sulfathiazole	78.6 ± 3	17.3 ± 7.3	41.6 ± 1.4	40.2 ± 2.5	46.4 ± 3.3	35.6 ± 5.5
Sulfamerazine	94.3 ± 5.6	89.2 ± 13.1	67.5 ± 10	82.3 ± 13.2	86.2 ± 25.1	57.4 ± 16
Sulfafurazole	75.4 ± 64	18.5 ± 30.9	39.8 ± 3.9	47.9 ± 0.9	37.9 ± 0.9	23.3 ± 3.6
Sulfadimethoxine	92.3 ± 10	84 ± 55.4	84.1 ± 10.4	91.2 ± 16.3	90.3 ± 12.1	87.6 ± 18.3
Sulfadimoxine	93.6 ± 2.9	81.3 ± 35.7	76.8 ± 4.8	72.4 ± 3.8	62.4 ± 4.1	43.1 ± 8.1
Sulfamethizole	62.7 ± 8.6	35 ± 22.9	36.2 ± 8.8	79.9 ± 8.2	46.2 ± 3.9	71.6 ± 14
Sulfabenzamide	88.1 ± 3.4	42.1 ± 6.2	22.6 ± 10.1	23.1 ± 0.1	78.1 ± 3.4	93.1 ± 4.5
Sulfisomidine	65.3 ± 4.2	65 ± 16.2	0.3 ± 0.3	72.1 ± 6.9	52.1 ± 6.9	75.2 ± 15.3
Sulfamethazine	98.1 ± 14.8	44.9 ± 16.4	78.1 ± 17.3	70.5 ± 1.6	86.2 ± 25.1	70.5 ± 9
Sulfametoxydiazine	67.2 ± 12.4	55.1 ± 29.7	5.1 ± 1.1	44.2 ± 6.9	66.7 ± 9.9	55.3 ± 3.8
Sulfamethoxypyridazine	92.1 ± 9.2	77.2 ± 18.5	32.6 ± 21	44.8 ± 13.2	82.1 ± 9	25.2 ± 14.9
Sulfamonomethoxine	87.2 ± 10.4	82.5 ± 21.2	57 ± 15.2	57.5 ± 36.3	88.3 ± 15.2	62.8 ± 8.8
Sulfachloropyridazine	78 ± 8.1	23.2 ± 21.4	65.1 ± 24.9	74 ± 27.5	55.1 ± 4.4	40.3 ± 20.3
Sulfachloropyrazine	75.5 ± 9.8	32.3 ± 23.2	56.9 ± 16.2	64.2 ± 1.7	67.3 ± 5.7	77.9 ± 35.7
Sulfaquinoxaline	85.6 ± 15.8	88.5 ± 25.3	29.4 ± 11.8	81.7 ± 24.2	90.7 ± 13.5	74.7 ± 21.3
Sulfanitran	95.2 ± 41	96.1 ± 1.9	93.7 ± 4.0	93.4 ± 21.6	93.4 ± 2.6	99.8 ± 3.2
Sulfisoxazole	75.1 ± 8.6	45.1 ± 27.6	32.2 ± 13.2	52.6 ± 8.5	36.6 ± 1.5	44.4 ± 13.6
Sulfaphenazole	68.3 ± 12.9	27.6 ± 39.8	18.4 ± 1.3	22.7 ± 0.5	25.4 ± 3.5	11.8 ± 8.7
Sulfapyrazole	67.5 ± 11.2	9.7 ± 2.4	31 ± 0.4	34.4 ± 11.3	19.7 ± 3.4	35.8 ± 4.8

**Table 2 molecules-27-04953-t002:** Linear, LOD and LOQ of the developed method.

Drugs	Linear Equations	Concentration Range (ng/mL)	R	LOD (μg/g)	LOQ (μg/g)
Sulfadiazine	Y = 0.0066 + 8.086X	1–100	0.9961	0.6	2
Sulfamethoxazole	Y = −0.0077 + 6.813X	1–100	0.9972	0.6	2
Sulfathiazole	Y = 0.0068 + 6.82X	1–100	0.9956	0.7	2
Sulfamerazine	Y = −0.0001 + 12.39X	0.5–200	0.9973	0.3	1
Sulfafurazole	Y = −0.0023 + 13.38X	1–100	0.9985	0.3	1
Sulfadimethoxine	Y = 0.0083 + 1.883X	1–100	0.9977	0.3	1
Sulfadimoxine	Y = 0.0066 + 16.923X	1–100	0.9981	0.3	1
Sulfamethizole	Y = 0.0006 + 10.667X	1–100	0.9964	0.2	0.6
Sulfabenzamide	Y = −0.0096 + 1.762X	1–100	0.9976	0.3	1
Sulfisomidine	Y = −0.0027 + 6.626X	0.5–50	0.9974	0.1	0.3
Sulfamethazine	Y = 0.0082 + 7.666X	0.5–50	0.9973	0.1	0.3
Sulfametoxydiazine	Y = −0.0081 + 6.697X	1–100	0.9967	0.3	1
Sulfamethoxypyridazine	Y = −0.0019 + 9.768X	0.5–50	0.9968	0.3	1
Sulfamonomethoxine	Y = 0.0021 + 2.81X	0.5–50	0.9991	0.2	0.6
Sulfachloropyridazine	Y = −0.0006 + 3.976X	1–100	0.9991	0.3	1
Sulfachloropyrazine	Y = −0.0013 + 7.713X	1–100	0.9979	0.3	1
Sulfaquinoxaline	Y = −0.0063 + 6.301X	1–100	0.9974	1.5	5
Sulfanitran	Y = 0.0076 + 13.006X	2–200	1.000	1.5	5
Sulfisoxazole	Y = 0.0003 + 12.186X	1–100	0.9986	0.7	2
Sulfaphenazole	Y = −0.0068 + 11.336X	1–100	0.9956	0.2	0.6
Sulfapyrazole	Y = 0.0003 + 6.61X	1–100	0.9992	0.2	0.6

**Table 3 molecules-27-04953-t003:** Recovery of within-day and between-day for poultry eggs (n = 6).

Drugs	Within-Day (Recovery ± RSD%)			Between-Day (Recovery ± RSD%)		
	5 (μg/kg)	10 (μg/kg)	100 (μg/kg)	5 (μg/kg)	10(μg/kg)	100 (μg/kg)
Sulfadiazine	89 ± 11.7	91.5 ± 14.4	84.9 ± 17.9	86.3 ± 5.5	92.4 ± 7.4	91 ± 11.7
Sulfamethoxazole	89 ± 4.3	86 ± 5.1	86.6 ± 14.8	97.6 ± 7.2	91.5 ± 10.9	86.9 ± 8.1
Sulfathiazole	88.9 ± 12.5	94.5 ± 8.2	95.7 ± 13.6	90.3 ± 3.2	98.3 ± 9	103.4 ± 8.4
Sulfamerazine	93.6 ± 1.9	97.6 ± 1.7	93 ± 8.7	93.2 ± 7.6	103.6 ± 12.4	96.9 ± 2.0
Sulfafurazole	93.6 ± 9.7	92.6 ± 11.5	94.5 ± 11.6	92 ± 11.2	92.6 ± 12	99 ± 13.4
Sulfadimethoxine	86.9 ± 7.2	89.6 ± 7.5	86.2 ± 9.3	99.5 ± 12.1	89 ± 0.6	92.8 ± 4.7
Sulfadimoxine	90.8 ± 13.5	102 ± 1.2	88.9 ± 13.6	86.2 ± 14.6	95.9 ± 0.4	85.9 ± 4.6
Sulfamethizole	100.1 ± 3.6	87.4 ± 4.8	93.7 ± 9.6	105.8 ± 4.8	90.3 ± 10.1	102.9 ± 9.5
Sulfabenzamide	102.3 ± 13.1	89.4 ± 12.6	99.3 ± 7	93.1 ± 6.2	95 ± 12.4	103.7 ± 10.1
Sulfisomidine	93.3 ± 6.9	104.5 ± 1.2	102.8 ± 10.1	101 ± 9.2	100.4 ± 3.1	94.1 ± 4.7
Sulfamethazine	97.2 ± 1.4	99.7 ± 2.7	103.7 ± 9.6	90 ± 4.6	104.9 ± 10.8	93.1 ± 10.5
Sulfametoxydiazine	84.3 ± 6.9	94.7 ± 8	93.6 ± 10.6	88 ± 11.8	85.9 ± 6.8	89.4 ± 12
Sulfamethoxypyridazine	87.5 ± 9.8	89.7 ± 12.5	90.9 ± 12.9	89.1 ± 4.2	89.3 ± 2.9	88.2 ± 8.8
Sulfamonomethoxine	86.1 ± 1.7	103.5 ± 9.9	95.9 ± 10.8	98.2 ± 0.2	100.1 ± 9.4	98.3 ± 10.6
Sulfachloropyridazine	94.8 ± 8.5	87.8 ± 7.7	96.9 ± 12.3	87.7 ± 10.2	87.5 ± 13.4	94.2 ± 7.9
Sulfachloropyrazine	103.2 ± 6.5	96.6 ± 11.8	93 ± 15.3	88 ± 11.6	93.5 ± 8.6	86.9 ± 13.4
Sulfaquinoxaline	101.1 ± 2.5	103.7 ± 8.7	90.4 ± 10.5	103.8 ± 8.5	100.2 ± 12.2	96.6 ± 14.9
Sulfanitran	101.4 ± 10.1	102.2 ± 12.8	96.9 ± 1.8	92.5 ± 7.6	102.7 ± 13.7	94.1 ± 11.7
Sulfisoxazole	99.5 ± 3.3	90.7 ± 14	103.5 ± 11.3	93.3 ± 7.9	85.6 ± 10.6	104 ± 1.6
Sulfaphenazole	104.2 ± 12.5	90.1 ± 9.3	87.7 ± 12.9	103.1 ± 6.6	89.7 ± 4.9	95.7 ± 14.7
Sulfapyrazole	103.1 ± 9	101.3 ± 2.8	88.7 ± 9.9	99.7 ± 7.8	96.5 ± 3.4	88.2 ± 13.6

**Table 4 molecules-27-04953-t004:** Comparison of the proposed method with other reported methods.

Kind of Drugs	Extraction Solvent	Purification	LOD (μg/kg)	Reference
14	20 mM phosphate solution	Polymer monolith microextraction	0.9–9.8	[35] Zheng, 2008
9	Sodium succinate buffer	SPE (HLB)	10–50	[14]
12	Hot water	Matrix solid-phase dispersion	2–6	[41]
7	ACN	Magnetic-multiwall carbon nanotubes as adsorbents	1.4–2.8	[13]
16	Ethanol	Immunoaffinity column	3.0	[15]
16	Phosphate solution	SPE (MCX)	0.5~1	[16]
21	Phosphate solution	SPE (MIP)	0.1~1.5	This work

## Data Availability

Not applicable.

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
