# Peer review of "Simultaneous Determination of 21 Sulfonamides in Poultry Eggs Using Ionic Liquid-Modified Molecularly Imprinted Polymer SPE and UPLC–MS/MS"

_molecules, 2022, doi:10.3390/molecules27154953_

Round 1
Reviewer 1 Report
This manuscript demonstrates the use of sulfamethazine as template molecule, methyl acrylic acid and selected ionic liquids (1-aminopropyl-3-methylimidazolium bromide, 1-allyl-3-vinylimidazolium bromide, butyl-3-methylimidazole bromide and 2,1-ethyl-3-methylimidazole bromide) as functional monomers for the synthesis of ionic liquid-modified molecularly imprinted polymer. The polymers were characterized by FTIR and SEM, as well as evaluated by means of adsorption rate. The SPE purification procedures were optimized with the selected pH of phosphate buffer solution (3.0-10.0), amount of polymer added (20-500mg), and the type (ACN, MeOH: acetic acid with different ratios) and volume (1-5mL) of elution solvent. The extraction performance was examined on 21 sulfonamides present in egg samples.
Comments:
- Page 1 Lines 2 to 5 – Title too wordy.
- Page 2 Lines 43 to 48 – Does the author trying to tell, there are 13 research that could not extract sulfonamides from the egg samples with good recoveries?
- Page 3 Lines 114 to 116 – These statements were found not clear and require further clarifications.
- Page 3 Lines 117 to 118 – What is MMIP stand for?
- Page 3 Line 125 – PBS?
- Page 4 Lines 157 to 159 – “AMB” is not found in Figure S1, kindly state properly what is MMIP and the meaning of “value of sorbent”?
- Page 5 Figure 2 – Show the scaling clearly.
- Page 5 Lines 179 to 181 – What is the difference between adsorption effect and adsorption rate/property?
- Page 5 Table 1 – Since MMIP was the most efficient one as written in Section 3.1 (Page 4 Line 151), why 6 MIPs are compared again? If 6 MIPs will be compared in terms of adsorption rate, why does the “Selection of IL” in Section 3.1 need to be conducted? What is the purpose of “Selection of IL”?
- Page 6 Line 205 to 206 – “Four types of IL-MIPs (i.e., VMIP [14], BMIP [11], and EMIP [13]) …” Where is the fourth one?
- Page 7 Table 2 – R of sulfamerazine was not exceeding 0.99? The concentration range used in calibration curves was smaller than LOQ? Is the unit for LOD and LOQ are correctly used?
- Page 8 Table 3 – There is a data showing %RSD of more than 15%. Clarification required at this point.
- Page 9 Line 254 and Line 259 – Repeating Sentences.
- Page 9 Line 257 – What is acid ME?
- Page 9 Table 4 – What is the unit for LOD?
- Page 10 Line 284 – Kindly check if the unit used was correct.
- Page 10 Figure 3 – Figure 3 is difficult to understand. Which three were the positive egg samples from the animal study? Stability test also conducted? It was not stated in the paragraph above.
- Page 10 Line 290 – The first and last “EGMA” was written. Kindly check again if it is EGMA or EDMA.
- Table S2 – In the caption, 21 sulfonamides or 27 sulfonamides? Included internal standards?
- Figure S1 – What is y-axis? No caption?
- Figure S2 – No caption?
- Figure S3 – What is y-axis? No caption?
- Figure S4 – No caption and axis labels?
- Last but not least, the manuscript requires major language polishing.
Author Response
- Page 1 Lines 2 to 5 – Title too wordy.
Response 1::Title have been shorten
- Page 2 Lines 43 to 48 – Does the author trying to tell, there are 13 research that could not extract sulfonamides from the egg samples with good recoveries?
Response 2: no, many research have good recoveries for SAs in egg, However, these methods have no specificity for SAs, and usually extract some impurities, which will affect the determination of sulfonamides. Therefore, additional time and pretreatment steps are required to remove interference. the section have been revised.
- Page 3 Lines 114 to 116 – These statements were found not clear and require further clarifications.
Response 3:These statements have been revised In order to clarify the experimental steps
- Page 3 Lines 117 to 118 – What is MMIP stand for?
Response 3: MMIP is an IL-MIP with 1-aminopropyl-3-methylimidazole bromide (AMB) as monomer, the definition of MMIP have been added.
- Page 3 Line 125 – PBS?
Response 4: phosphate buffer saline, revised.
- Page 4 Lines 157 to 159 – “AMB” is not found in Figure S1, kindly state properly what is MMIP and the meaning of “value of sorbent”?
Response 5: MMIP is an IL-MIP with 1-aminopropyl-3-methylimidazole bromide (AMB) as monomer
- Page 5 Figure 2 – Show the scaling clearly.
Response 6: the scaling have been shown clearly.
Page 5 Lines 179 to 181 – What is the difference between adsorption effect and adsorption rate/property?
Response 7: these scanning electron microscopy of all mip only proved MMIP have been Successfully synthesized, which are unable to clarify differences between adsorption effect and adsorption rate/property
- Page 5 Table 1 – Since MMIP was the most efficient one as written in Section 3.1 (Page 4 Line 151), why 6 MIPs are compared again? If 6 MIPs will be compared in terms of adsorption rate, why does the “Selection of IL” in Section 3.1 need to be conducted? What is the purpose of “Selection of IL”?
Response 8: Table 1 only proves that MMIP was the most effective for SMZ, but there is no result for the adsorption effect of other SA, therefore, 6 MIPs are compared with adsorption properties 21 SAs. The results further proved advantage of selected IL. There are added to section
- Page 6 Line 205 to 206 – “Four types of IL-MIPs (i.e., VMIP [14], BMIP [11], and EMIP [13]) …” Where is the fourth one?
Response 9: it is MMIP, and added.
- Page 7 Table 2 – R of sulfamerazine was not exceeding 0.99? The concentration range used in calibration curves was smaller than LOQ? Is the unit for LOD and LOQ are correctly used?
Response 10: there are input error and revised. The calculation of LOQ and LOD takes into account the dilution and constant volume of the experimental process of eggs. Therefore, The concentration range used in calibration curves was smaller than LOQ. the unit are correct.
- Page 8 Table 3 – There is a data showing %RSD of more than 15%. Clarification required at this point.
Response 11: there are input error and revised.
- Page 9 Line 254 and Line 259 – Repeating Sentences.
Response 12: There have revised
- Page 9 Line 257 – What is acid ME?
Response 13:MeOH + acetic acid (9:1, V/V), have revised.
- Page 9 Table 4 – What is the unit for LOD?
- Page 10 Line 284 – Kindly check if the unit used was correct.
Response 14: there have checked and added.
- Page 10 Figure 3 – Figure 3 is difficult to understand. Which three were the positive egg samples from the animal study? Stability test also conducted? It was not stated in the paragraph above.
Response 15: SMZ withdrawal 3,7,14 is egg samples from 3,7,14 days after drug withdrawal.Positive results were analysed in accordance with the standard method of China and this method ,therefore Stability test have no conducted
- Page 10 Line 290 – The first and last “EGMA” was written. Kindly check again if it is EGMA or EDMA.
Response 16: EDMA IS EGMA, have revised.
- Table S2 – In the caption, 21 sulfonamides or 27 sulfonamides? Included internal standards?
Response1 7: 21 sulfonamides have revised.
- Figure S1 – What is y-axis? No caption?
Response 18: y-axis and caption have added.
- Figure S2 – No caption?
Response 19: caption have added.
- Figure S3 – What is y-axis? No caption?
Response 20: y-axis and caption have added.
- Figure S4 – No caption and axis labels?
Response 21: y-axis and caption have added.
- Last but not least, the manuscript requires major language polishing.
Response 22: language have revised with editor.
Reviewer 2 Report
This study presents simultaneous determination of twenty-one sulfonamides in eggs using ionic liquid-modified molecularly imprinted polymer solid-phase extraction by ultraperformance LC–MS/MS (UPLC– MS/MS). The proposed method proved to be sensitive and showed good recoveries of the studied sulfonamides in spiked egg samples. The experiments are well- described and the results support their conclusions. This study is novel and fits in the scope of Molecules. I accept this paper as such and recommend its publication.
Author Response
English language and style have been revised.
Reviewer 3 Report
The manuscript is interesting, however improvements are necessary.
- In the title, verify the spacing before … ultraperformance.
- In the title, specify the type of eggs: poultry eggs?
- In the title: why a hyphen to separate materi-al?
- Verify how to write the authors names.
- Keywords, insert one space after each semi-colon.
- Introduction section and in the whole manuscript, when you write the references number, insert one space before the first bracket (see line 29, 31….).
- Line 44, delete one dot before… However.
- Line 68 and in the whole manuscript, do not use superscript to indicate the references.
- 2.1 sub-section, line 101 and in the whole manuscript: what type of water?
- 2.2.3 sub-section, line 138: 150 mm length (to be consistent with other parameters).
- Caption of Table 1 and all tables captions: insert one dot after the caption number. Use a recently published paper as a template.
- Table 1, insert one space after adsorption rates.
- Table 1, sometime you have written the name of molecules in capital letter and sometime in small letter. Please, be consistent here and in all tables and figures.
- Table 1, column 2 from right: re-arrange the table and try to do not write the decimal in a different line.
- Table 4, column 1 form right: be consistent with spacing and punctuation.
- 3.7 sub-section, lines 275-276-277, verify the wording: 2019)Error! Bookmark not defined.
- Figure 3, y axis (you have used a fraction), line 99 you have used an exponent. Please, be consistent in the whole manuscript when you indicate a quantity.
- Page 10, separate the conclusions section by Funding. Please, use some recently published paper as a template.
- The references section is not arranged as per the instructions for authors of Molecules.
- References section: verify the spacing between words, symbols, numbers, there is a lot of inaccuracies.
- The references section has to be arranged as required by Molecules (MDPI).
- Insert the captions of tables and figures in the supplementary material file.
Author Response
The manuscript is interesting, however improvements are necessary.
- In the title, verify the spacing before … ultraperformance.
Response 1: have been revised to UPLC.
- In the title, specify the type of eggs: poultry eggs?
Response 2: poultry eggs, added
- In the title: why a hyphen to separate materi-al?
Response 3: hyphen have been deleted
- Verify how to write the authors names.
Response 4 :the authors names have been revised.
- Keywords, insert one space after each semi-colon.
Response 5: there are revised.
- Introduction section and in the whole manuscript, when you write the references number, insert one space before the first bracket (see line 29, 31….).
Response 6: there are revised.
- Line 44, delete one dot before… However.
Response 7: there have been deleted.
- Line 68 and in the whole manuscript, do not use superscript to indicate the references.
Response 8: there have been revised.
- 2.1 sub-section, line 101 and in the whole manuscript: what type of water?
Response 9: Water was purified using the Milli-Q water purification system from Millipore, which have been mentioned on 2.1. Materials and reagents.
- 2.2.3 sub-section, line 138: 150 mm length (to be consistent with other parameters).
Response 10: length have been added.
- Caption of Table 1 and all tables captions: insert one dot after the caption number. Use a recently published paper as a template.
Response 11::there have revised refer with a recently published paper as a template.
- Table 1, insert one space after adsorption rates.
Response 12:revised.
- Table 1, sometime you have written the name of molecules in capital letter and sometime in small letter. Please, be consistent here and in all tables and figures.
Response 13: capital letter and small letter have been consistent
- Table 1, column 2 from right: re-arrange the table and try to do not write the decimal in a different line.
Response 14:revised.
- Table 4, column 1 form right: be consistent with spacing and punctuation.
Response 15:revised.
- 3.7 sub-section, lines 275-276-277, verify the wording: 2019)Error! Bookmark not defined.
Response 16:revised.
- Figure 3, y axis (you have used a fraction), line 99 you have used an exponent. Please, be consistent in the whole manuscript when you indicate a quantity.
Response 17: The eggs collected 14 days after drug withdrawal were 1.8 ± 0.3 µ g/ kg, which is difficult to express different with other reuslt if using the proportional axis. herefore ,therefore,y axis using logarithmic scale better shows result for reader to understand
- Page 10, separate the conclusions section by Funding. Please, use some recently published paper as a template.
Response 18:revised.
- The references section is not arranged as per the instructions for authors of Molecules.
- References section: verify the spacing between words, symbols, numbers, there is a lot of inaccuracies.
- The references section has to be arranged as required by Molecules (MDPI).
Response 19:The references have been revised reference with requirement of Molecules
- Insert the captions of tables and figures in the supplementary material file.
Response 20:tables and figures in the supplementary material file have been revised.
Round 2
Reviewer 1 Report
The Authors have addressed all of my concerns with the original manuscript. The revised manuscript is ready for publication.
Reviewer 3 Report
Authors have adjusted the manuscript as requested. Furthermore, they have correctly replayed to my observation. The manuscript in present status could be publish.